# Effect of Growth Factors and Hormones during In Vitro Growth Culture of Cumulus-Oocyte-Complexes Derived from Small Antral Follicles in Pigs

**DOI:** 10.3390/ani13071206

**Published:** 2023-03-30

**Authors:** Minji Kim, Ji-Eun Park, Yongjin Lee, Seung-Tae Lee, Geun-Shik Lee, Sang-Hwan Hyun, Eunsong Lee, Joohyeong Lee

**Affiliations:** 1Laboratory of Theriogenology, College of Veterinary Medicine, Kangwon National University, Chuncheon 24341, Gangwon, Republic of Korea; 2Division of Applied Animal Science, College of Animal Life Science, Kangwon National University, Chuncheon 24341, Gangwon, Republic of Korea; 3Laboratory of Inflammatory Diseases, College of Veterinary Medicine, Kangwon National University, Chuncheon 24341, Gangwon, Republic of Korea; 4Laboratory of Veterinary Embryology and Biotechnology (VETEMBIO), College of Veterinary Medicine, Chungbuk National University and Institute of Stem Cell & Regenerative Medicine, Chungbuk National University, Cheongju 28644, Chungbuk, Republic of Korea; 5Institute of Veterinary Science, Kangwon National University, Chuncheon 24341, Gangwon, Republic of Korea

**Keywords:** growth factor, insulin, small antral follicle, in vitro growth culture, pig

## Abstract

**Simple Summary:**

This study investigated the effect of various growth factors and hormones in an in vitro growth medium on in vitro maturation and the developmental competence of oocytes derived from small antral follicles in pigs. Cumulus–oocyte complexes derived from small antral follicles (<3 mm) were cultured in an in vitro growth medium supplemented with epidermal growth factor (EGF), insulin-like factor-1 (IGF-1), insulin, or growth hormone (GH). The effects on oocyte growth, nuclear progression, oocyte maturation, and embryonic development after parthenogenesis were examined. Insulin treatment during oocyte growth was found to have a positive effect on nuclear maturation, embryonic development after parthenogenesis, and the expansion of cumulus cells. Insulin showed positive results by enhancing cytoplasmic maturation, decreasing free radical content, and increasing maturation-promoting factor activity.

**Abstract:**

This study evaluated the effect of various growth factors and hormones in an in vitro growth (IVG) medium on the in vitro maturation (IVM) and developmental competence of oocytes derived from small antral follicles (SAFs) in pigs. Cumulus–oocyte complexes (COCs) derived from SAFs were either untreated or treated with epidermal growth factor (EGF), insulin-like factor-1 (IGF-1), insulin, or growth hormone (GH) for 2 days of IVG. Following IVG, COCs were cultured for maturation, and IVM oocytes were induced for parthenogenesis (PA). During IVG, the nuclear maturation of oocytes was significantly increased by the insulin treatment compared to other treatments. Moreover, the insulin treatment significantly increased blastocyst formation after PA relative to the No-IVG, control, EGF, and GH treatments. The cumulus expansion score after IVG-IVM was significantly higher in the insulin group than in the other groups. The glutathione (GSH) contents in IVM oocytes were increased through treatment with IGF, insulin, and GH compared to those of No-IVG oocytes. The level of reactive oxygen species (ROS) in IVM oocytes in all treatment groups was significantly lower after IVG culture than in the No-IVG group. The maturation-promoting factor (MPF) activity after IVM in the insulin-treated oocytes was significantly higher than that of the oocytes treated with EGF, IGF-1, and GH. In conclusion, this study demonstrates that insulin treatment during IVG culture improves the maturational and developmental competence of oocytes derived from SAFs in pigs through its effect on cumulus cell expansion and cytoplasmic microenvironments, such as GSH, ROS, and MPF activity.

## 1. Introduction

Pigs are widely used as animal models for human disease and organ transplantation because of their many anatomical, genetic, and physiological similarities to humans [1,2]. To date, breeding efficiency in domestic animal species using assisted reproductive technologies such as in vitro maturation (IVM) and in vitro fertilization (IVF) is generally lower than with natural breeding. Accordingly, various studies have been performed to improve the developmental competence of porcine oocytes by modifying the culture system for oocyte maturation and embryonic development [3,4].

In pig reproduction, immature oocytes derived from small antral follicles (SAFs) less than 3 mm in diameter show lower maturational and developmental competence than oocytes derived from medium antral follicles (MAFs) larger than 3 mm in diameter [5,6]. Thus, cumulus-oocyte-complexes (COCs) collected only from MAFs have been used for the in vitro production (IVP) of embryos after IVM, IVF, or somatic cell nuclear transfer (SCNT), whereas, especially in pigs, many SAF-derived COCs (SAFCOCs) have not been used. Consequently, more than half of the oocytes contained in the superficial follicles are excluded from the IVP of porcine embryos. If the developmental potential of SAFCOCs could be improved, it would be possible to secure a greater number of oocytes as valuable genetic resources for the IVP of porcine embryos. Recently, various studies have been conducted to establish an efficient in vitro growth (IVG) culture system in which SAFCOCs are induced to grow without precocious nuclear maturation, allowing time for sufficient cytoplasmic maturity. Cyclic adenosine monophosphate (cAMP) plays an important role in inhibiting meiotic resumption [7]. Furthermore, maintaining gap junction communication between oocytes and cumulus cells during IVM of immature oocytes is critical for oocyte cytoplasmic maturation [8]. In pigs, the IVG culture system, using a medium containing dibutyryl cAMP (dbcAMP), induces oocyte growth while inhibiting spontaneous meiotic resumption, and has been widely used to produce mature oocytes. These studies reported improved developmental competence of SAFCOCs comparable to that of MAFs-derived COCs (MAFCOCs) [6,9,10].

This study was designed for the IVG of small follicle-derived COCs based on effective substances and optimal concentrations in the IVM process of medium-sized follicle-derived COCs. This suggests a possibility of further improvement of developmental competence by optimizing an IVG culture system after modification of medium components. The growth and development of follicles, oocytes, and embryos depend on the continuous supply of hormones and growth factors [11]. Growth factors such as epidermal growth factor (EGF) and insulin-like growth factor-1 (IGF-1) are found in the ovary and affect ovarian cell function. Porcine follicular fluid has a concentration of 10–20 ng/mL EGF [12]. EGF stimulates nuclear maturation during the IVM of mouse [13], rat [14], pig [15], and sheep [16] oocytes. EGF also promotes the expansion of cumulus cells and improves cytoplasmic maturation as well as nuclear maturation [17]. IGF-1 is an essential component of ovarian mechanisms involved in the regulation of follicular growth and corpus luteum function [18]. IGF-1 stimulates and enhances the maturation of pig [19] and sheep [16] oocytes in vitro. Oberlender et al. reported that follicular fluid derived from small follicles (2–5 mm) contained a lower concentration of IGF-1 compared to follicular fluid derived from large follicles (6–10 mm) [20]. Insulin regulates ovarian development and oocyte production [21]. In vitro, insulin stimulates the proliferation and steroid production of granulosa and theca cells [22]. Growth hormone (GH) plays an important role in controlling follicle development and oocyte maturation [23]. It has been reported that GH acts directly on ovarian follicles and that the action of GH depends on the size of the follicle, showing more effects on cells of large follicles than on those of small-sized follicles [24]. The IVM of MAFCOCs in GH-supplemented cultures increases the maturation rate but does not affect fertilization or embryonic development [25].

During the oocyte maturation process, there is an interaction between the oocytes and the cumulus cells surrounding the oocytes. It is well known that cumulus cells affect oocyte maturation and fertilization, along with enhancing cytoplasmic maturation, which is essential for oocytes to support normal embryonic development after fertilization [26]. It is possible that various growth factors are involved in the proliferation and differentiation of cumulus cells [27], as well as in oocyte maturation itself. Therefore, treatment with EGF, IGF-1, insulin, and GH during the IVM of MAFCOCs has been shown to positively affect oocyte maturation and early embryonic development. However, the effects of such treatment during IVG culture on the maturation and embryonic developmental competence of SAFCOCs have not been fully studied.

Therefore, it was hypothesized in this study that treatment of SAFCOCs with individual growth factors or hormones during IVG culture would influence oocyte growth and result in later embryonic development in pigs. To verify this hypothesis, SAFCOCs were cultured in an IVG medium supplemented with EGF, IGF-1, insulin, or GH, and the effects on oocyte size, nuclear progression, oocyte maturation, and embryonic development after parthenogenetic activation (PA) were examined.

## 2. Materials and Methods

### 2.1. Ethics Statement

The animal study was reviewed and approved by the Committee on Ethics of Animal Experiments of the Chungbuk National University (Permit Number: CBNUA-1733-22-01).

### 2.2. Culture Media and Chemicals

All chemicals used in this study were purchased from Sigma-Aldrich (St. Louis, MO, USA) unless specified otherwise. The base medium for the IVG culture for SAFCOCs was Minimum Essential Medium alpha medium (a-MEM; Invitrogen, Carlsbad, CA, USA) supplemented with 0.1% (*w*/*v*) polyvinyl alcohol (PVA), 0.4 mM pyruvate, and 75 μg/mL kanamycin. For the IVM of immature porcine oocytes, medium-199 (M-199) (Invitrogen, Grand Island, NY, USA) was used. This IVM medium was supplemented further with 10% (*v*/*v*) porcine follicular fluid, 0.4 mM pyruvate, 0.6 mM cysteine, 10 ng/mL epidermal growth factor (EGF), 1 μg/mL insulin, and 75 μg/mL kanamycin. The medium for the in vitro culture (IVC) of PA embryos was porcine zygote medium (PZM)-3, which contained 0.3% (*w*/*v*) bovine serum albumin (BSA). The IVC medium was modified in this study by adding 0.34 mM tri-sodium citrate, 2.77 mM myo-inositol, and 10 μM β-mercaptoethanol [28].

### 2.3. Oocyte Collection and IVG of SAF-Derived Oocytes

The ovaries of pre-pubertal gilts were obtained at a local slaughterhouse and transported in sterile saline at 36–38 °C within 2 h. Immature COCs from these ovaries were collected from small antral follicles of less than 3 mm in diameter by aspirating them using a disposable needle attached to a 10 mL syringe. The SAFCOCs with multiple layers of cumulus cells were selected and washed in HEPES-buffered Tyrode’s medium containing 0.05% (*w*/*v*) PVA (TLH-PVA). The optimal culture period for IVG was determined by our group’s other experimental results (unpublished data). Forty to fifty SAFCOCs were cultured for 2 days (48 h) in 500 μL of IVG medium supplemented with 1 mM of dbcAMP in a 4-well multidish (Nunc, Roskilde, Denmark) at 39 °C under a humidified atmosphere of 5% CO_2_ and 95% air. To elucidate the effects of growth factors and hormones, the base IVG medium was supplemented with either 10 ng/mL EGF, 50 ng/mL IGF-1, 1 μg/mL insulin, 100 ng/mL GH, or nothing (“None”), according to the experimental design.

### 2.4. Experimental Design

We performed a series of experiments to determine the effect of these growth factors and hormones on various developmental outcomes. In Experiment 1, the outcomes we focused on were oocyte maturation after IVM and embryonic development after PA. In Experiment 2, we observed the cumulus expansion and the growth of SAFCOCs. In Experiment 3, we examined the cytoplasmic maturation by measuring the intra-oocyte reactive oxygen species (ROS) and glutathione (GSH) contents of the IVM oocytes. In Experiments 4 and 5, we observed the nuclear progression of SAFCOCs after IVG and IVM, respectively. In Experiment 6, we examined cytoplasmic maturation by assessing the maturation-promoting factor (MPF) activity of IVM oocytes. Finally, in Experiment 7, oocyte maturation and embryonic development after PA of SAF-derived oocytes that were treated with insulin during IVG were compared to those of MAF-derived oocytes.

### 2.5. IVM of IVG Oocytes

After IVG, the SAFCOCs were placed into each well of a 4-well culture dish containing 500 μL of IVM medium enriched with 80 μg/mL FSH (Antrin R-10; Kyoritsu Seiyaku, Tokyo, Japan) and 10 IU/mL human chorionic gonadotrophin (Intervet International BV, Boxmeer, Holland). The SAFCOCs were cultured at 39 °C under a humidified atmosphere of 5% CO_2_ and 95% air. After 22 h in the maturation culture, the SAFCOCs were washed three times in IVM medium and then cultured in a hormone-free IVM medium for an additional 22 h.

### 2.6. Examination of Cumulus Expansion and Nuclear Status of Oocytes

After IVM, cumulus cell expansion was assessed subjectively, as previously described [29]. The cumulus expansion scores were calculated for individual COCs. Briefly, no response was scored as zero. A minimum observable response, with the cells in the outermost layer of the cumulus becoming round and glistening as one. The expansion of outer cumulus cell layers is two. The expansion of all cumulus cell layers except the corona radiata is three. Additionally, the expansion of all cumulus cell layers including the corona radiata is four. After IVG or IVM, to assess the nuclear status of oocytes, COCs were denuded, fixed, and permeabilized in Dulbecco’s phosphate-buffered saline (D-PBS; Invitrogen) supplemented with 3.7% (*w*/*v*) paraformaldehyde and 1% (*v*/*v*) Triton X-100 at 17–25 °C for 15 min. Permeabilized oocytes were then placed on glass slides and stained with 5 μg/mL Hoechst 33342 before being overlaid with coverslips. The oocytes were examined by fluorescence microscopy (TE-300; Nikon, Tokyo, Japan) after incubation overnight at 4 °C. The nuclear stages of oocytes were classified into germinal vesicles (GV), GV breakdown (GVBD), metaphase I (MI), anaphase I and telophase I (AI/TI), and metaphase II (MII) according to the morphologic criteria for the characterization of meiotic stages reported previously [30] and Appendix A.

### 2.7. Measurement of Oocyte Diameter and Intra-Oocyte GSH and ROS Contents

After IVG or IVM, mechanical pipetting was performed while the oocytes were kept in 0.1% hyaluronidase because of the sticky character of the cumulus cells. Images of denuded oocytes were recorded using a digital camera (DS-L3; Nikon, Tokyo, Japan) attached to an inverted microscope (TE-300; Nikon). The size of each part of the oocytes was measured using image analysis software (version 1.46r; National Institutes of Health, Bethesda, MD, USA), and the size of the oocytes was calculated as described previously [28] and in Appendix A. Moreover, intra-oocyte GSH and ROS contents of IVM oocytes were measured by the method previously described [31]. Briefly, CellTracker Blue CMF2HC (4-chloromethyl-6.8-difluoro-7-hydroxycoumarin; Invitrogen) and H2DCFDA (2′,7′-dichlorodihydro-fluorescein diacetate; Invitrogen) were used to detect intracellular GSH and ROS contents as a blue fluorescence and green fluorescence, respectively. A group of MII oocytes from each group was collected 44 h after IVM and incubated in the dark for 30 min in TLH–PVA that was supplemented with 10 μM CellTracker and 10 μM H2DCFDA. After incubation, oocytes were washed with D-PBS (Invitrogen), which contained 0.1% (*w*/*v*) PVA, placed into 2-μL droplets and observed for fluorescence under an epifluorescence microscope (TE-300; Nikon) with a UV filter (370 nm and 460 nm). Fluorescent images were recorded and saved as graphic files in the TIFF format. The fluorescence intensity of the oocytes was analyzed by ImageJ software and normalized to untreated control oocytes.

### 2.8. PA, Post-Activation Treatment and Embryo Culture

After IVM, oocytes were denuded from their cumulus cells by gentle pipetting in an IVM medium with 0.1% (*w*/*v*) hyaluronidase. Each oocyte was rolled with a mouth pipette under a microscope, and only oocytes that extruded the first polar body (PB) were selected and allocated to PA. These oocytes were placed in an electrode chamber and PA was induced by applying two pulses of 120 V/mm of direct current for 60 μsec in a 280 mM mannitol solution supplemented with 0.05 mM MgCl_2_ and 0.1 mM CaCl_2_. Then, activated oocytes were washed properly in an IVC medium, which contained 7.5 μg/mL cytochalasin B, and incubated for 4 h in the same medium. PA embryos were washed thrice, placed in 30 μL IVC-medium droplets under mineral oil, and cultured for 7 days at 39 °C in a humidified atmosphere of 5% CO_2_, 5% O_2_, and 90% N_2_. Cleavage and blastocyst formation (based on the number of PA embryos cultured) were evaluated on days 2 and 7, respectively, with the day of PA designated as day 0. The total blastocyst cell count was performed using Hoechst 33342 staining under an epifluorescence microscope. The number of cells in porcine morula ranges from 9 to 16, and the early blastocyst has more than 17 cells [32]. Only PA embryos having a blastocoel cavity with more than 17 nucleus when stained with Hoechst 33342 were recognized as blastocysts [33].

### 2.9. MPF Activity Assay

The MPF assay was performed according to the manufacturer’s protocol using a commercially available ELISA kit (Porcine Maturation promoting factor ELISA Kit, My BioSource, Southern California, CA, USA) designed for quantitative analysis of MPF for cytoplasmic MPF activity in matured oocytes. Samples were placed in D-PBS containing 0.1% (*w*/*v*) PVA and stored at −75 °C until analysis. Eighty oocytes per treatment group were collected and used for the assay. Samples mixed with buffer were incubated for 1 h with the MPF-HRP conjugate on a coated plate. After incubation, the contents of each well were removed and washed five times. Finally, the reaction was stopped by adding stop solution, and chromaticity was measured using a microplate reader equipped with a 450 nm filter. The MPF concentration of each sample was calculated by comparison with a standard curve.

### 2.10. Statistical Analyses

All statistical analyses were performed using SAS (Version 9.4; SAS Institute, Cary, NC, USA). Data were analyzed using a general linear model followed by the least significant difference mean separation procedure when treatments differed at *p* < 0.05. Percentage data were arcsine transformed before analysis to maintain homogeneity of variances. Results were expressed as the mean ± standard error of the mean (SEM).

## 3. Results

### 3.1. Effect of Growth Factor or Hormone Treatment during IVG on Oocyte Maturation and Embryonic Development after PA of SAFCOCs

For this experiment, a total of 1,513 SAFCOCs were used. The proportion of oocytes that degraded after IVG-IVM was significantly lower in the No-IVG (4.1 ± 1.5%) than in the EGF (11.6 ± 2.9%) treatments (Table 1). Nuclear maturation to the MII stage was significantly decreased in EGF-treated oocytes (41.0 ± 5.4%), whereas insulin-treated oocytes showed a significantly higher nuclear maturation (93.4 ± 1.0%) to the MII stage compared to No-IVG, None, IGF-1, and GH-treated oocytes (80.0 ± 2.2, 65.3 ± 3.3, 72.6 ± 4.2, 56.3 ± 3.1%, respectively). After PA, embryo cleavage was not altered by the IVG treatment with growth factors or hormones. However, blastocyst formation increased in response to insulin (69.3 ± 3.2%) relative to the No-IVG (27.5 ± 3.4%), None (55.8 ± 5.2%), EGF (46.9 ± 3.7%), and GH (54.4 ± 5.4%) treatments. The mean cell number per blastocyst was higher in the insulin (41.0 ± 1.1 cells) and None (39.7 ± 1.5 cells) groups than in the No-IVG groups (34.5 ± 1.7%).

### 3.2. Effect of Growth Factors and Hormone Treatment during IVG on Cumulus Cell Expansion and Diameter of Oocytes after IVG-IVM

The cumulus expansion score after IVM was significantly higher in oocytes that were treated with insulin during IVG (3.30 ± 0.06) than in oocytes in the No-IVG, None, IGF-1, and GH groups (1.27 ± 0.08 to 1.92 ± 0.08) (Table 2 and Figure 1). After IVG, the diameters of the oocytes increased significantly (120.3 ± 0.5 to 122.0 ± 0.4 μm) compared to those before IVG (118.9 ± 0.5 μm). The diameters of the oocytes became smaller after IVM in all the treatment groups, while IVG oocytes (112.8 ± 0.8 to 116.9 ± 1.0 μm) were larger in diameter compared to oocytes in the No-IVG group (110.2 ± 0.9 μm) (Table 2).

### 3.3. Effect of Growth Factor or Hormone Treatment during IVG on Intra-Oocyte Contents of GSH and ROS after IVM

The intra-oocyte GSH content of IVM oocytes was significantly higher in the IGF-1 (1.22 ± 0.07 pixels/oocyte), insulin (1.20 ± 0.07 pixels/oocyte), and GH (1.24 ± 0.07 pixels/oocyte) groups than in the No-IVG (1.00 ± 0.07 pixels/oocyte) group. Irrespective of treatment, oocytes cultured for IVG showed significantly lower levels of ROS content (0.67 ± 0.04 to 0.72 ± 0.04 pixels/oocyte) compared to No-IVG oocytes (1.00 ± 0.04 pixels/oocyte) (Table 3).

### 3.4. Nuclear Stage after IVG and at 22 h of IVM of Oocytes Treated with Growth Factors or Hormones during IVG Culture

Following IVG, the nuclear status of oocytes was assessed to determine the effects of growth factor or hormone treatment during IVG on meiotic progression. All oocytes were at the GV stage before IVG culture, and nuclear progression was not influenced by growth factor or hormone treatment during IVG (Table 4). After 22 h of IVM, a higher proportion of oocytes remained at the GV stage in the GH (70.3 ± 1.1%) group compared to the No-IVG (32.9 ± 1.9%) group. A higher proportion of No-IVG oocytes (54.7 ± 2.5%) were found in the MI stage as compared to GH oocytes (23.1 ± 0.1%). The proportions of IGF-1 and insulin oocytes (41.4 ± 12.0 and 45.4 ± 9.7%) in MI were not significantly different from those of the None, EGF, and GH groups (32.2 ± 9.3, 25.7 ± 12.4, and 23.1 ± 0.1%) (Table 5).

### 3.5. MPF Activity in Oocytes Treated with Various Growth Factors or Hormones during IVG

Compared to the EGF, IGF-1, and GH treatments, the insulin treatment during IVG significantly increased the MPF activity of IVM oocytes. There were no significant differences between the No-IVG, None, and insulin treatment groups (Figure 2).

### 3.6. Oocyte Maturation and Embryonic Development after Parthenogenesis of SAF-Derived Oocytes Treated with Insulin during IVG Compared to Oocytes Derived from MAFs

For this experiment, a total of 248 MAFCOCs and 209 SAFCOCs were used. As shown in Table 6, nuclear maturation to the MII stage was significantly lower in SAF-derived oocytes (80.1 ± 4.1%) compared to oocytes derived from MAF (94.3 ± 0.9%). Following PA, embryo cleavage was significantly decreased in SAF-derived oocytes (77.1 ± 5.1%), compared to MAF-derived oocytes (88.9 ± 2.9%). However, there was no significant difference between SAF-derived and MAF-derived oocytes with regard to the blastocyst rate or the total number of cells in blastocysts.

## 4. Discussion

Mammalian oocytes spontaneously resume meiosis when they are released from follicles [34,35]. Even after IVM culture for 44 h, precocious nuclear maturation with the first PB extrusion occurs before oocytes reach sufficient cytoplasmic maturation [36]. This is especially true for SAF-derived COCs, which display an inferior competence in embryonic development as well as a lower ability to achieve nuclear maturation compared to oocytes collected from MAFs in vitro, although they can reach the MII stage after a conventional IVM culture. Numerous studies have attempted to improve the developmental potential of SAFCOCs in pigs, and some positive results have been obtained [6,37]. In this study, when EGF, IGF-1, insulin, or GH were added to an IVG medium, the insulin treatment stood out as having a beneficial effect on the oocyte maturation and embryonic development of SAF-derived porcine COCs. Recent work has shown additional growth factors supplemented in combination with EGF and IGF during IVM to be beneficial (notably fibroblast growth factor 2 and leukemia inhibitory factor) not only in PA embryos but also in SCNT and IVF embryos [38,39]. However, growth factor and hormone requirements in the in vitro environment may vary depending on the developmental period and origin of the oocyte.

The diameter of mature oocytes has a significant correlation with embryonic developmental capacity [40,41,42]. In our results, the diameter of oocytes cultured in IVG increased compared to oocytes that did not undergo IVG. Although the diameter of the oocytes decreased after IVM, it was confirmed that the IVG-cultured oocytes were still larger in diameter than the oocytes that did not undergo IVG. We think that the oocytes that underwent IVG while suppressing nuclear maturation had a long time to reach cytoplasmic growth because they were cultured longer than those that did not undergo IVG.

In vivo, the ovarian follicular microenvironment and maternal signals mediated through granulosa and cumulus cells serve to promote oocyte growth, development, and progressive acquisition of oocyte developmental capacity [43]. Cumulus cells play an important role in the maturation of oocytes and protect them from harmful environments. Cumulus cell expansion in vitro also plays an important role in embryonic development [22,44]. Furthermore, it provides oocytes with various substances and nutrients from the outside through the cells of the oocytes [45]. In this study, insulin treatment during IVG did not induce cumulus expansion during IVG but did induce it during IVM. Insulin stimulates granulosa cell proliferation, progesterone production, and corpus luteal cell steroidogenesis in most mammals [46]. Therefore, it can be assumed that the insulin added during the IVG period acted as an essential factor for the development of SAF-derived oocytes. As shown in Figure 1, many expanded cumulus cells were separated from the oocyte and attached to the bottom in the case of the EGF-treated group after IVG. During the porcine IVM process, this phenomenon was also observed when PVA was used instead of follicular fluid in the culture medium. In this study, PVA was used in the culture medium for defining medium conditions during IVG. It can be assumed that the use of PVA during IVG was responsible for the separation between the expanded cumulus cells and oocytes after IVM. IGF-1 and GH-treated SAFCOCs also showed low cumulus cell expansion after IVM. A previous study reported that adding EGF to the IVM of oocytes induced the expansion of cumulus cells [47,48]. However, in our IVG culture conditions, the addition of EGF caused premature cumulus expansion and stimulated the separation of cumulus cells from oocytes. This likely caused the premature cessation of gap junction communication between cumulus cells and oocytes, which results in low rates of nuclear maturation and low developmental capacity. Insulin did not induce cumulus cell expansion during IVG, so the cumulus cells remained intact in the oocytes. Insulin-treated oocytes have a positive effect on cumulus cell expansion, as there are enough remaining cells to expand during IVM. It is likely through this enhanced cumulus cell expansion that insulin-treated oocytes result in a better nuclear maturation rate and embryonic development after PA.

After 2 days of IVG, the nuclear status was confirmed, and there was no significant difference in the percentage of oocytes staying in GV among all treatment groups. This means that nuclear maturation was suppressed during IVG. Previous studies have reported that adding 1 mM dbcAMP to cultures arrests porcine SAFCOCs in the GV phase for more than 5 days [37]. During the maturation of porcine oocytes, adding EGF does not affect the cAMP pathway [49]. Our results also show that EGF, insulin, IGF-1, and GH also acted independently of the role of dbcAMP during IVG. After 22 h of IVM, the nuclear stage was confirmed, and the percentage of GH-treated oocytes remaining in GV was higher than that of the No-IVG group. The proportion of oocytes that progressed to the MI stage showed the opposite result. It has been reported that GH receptors are localized in follicular cells in most species (rat [50], bovine [51], porcine [52]). Therefore, it seems that the addition of GH in the IVG culture could not have a positive effect on the growth and maturation of oocytes.

Depending on the meiosis phase of the oocytes, MPF activity either increases or decreases. Increased MPF in the oocyte cytoplasm leads to the maturation of oocytes at certain concentrations. MPF increases slowly before GVBD until reaching a certain concentration, at which point it remains until fertilization [53]. After IVG-IVM, only mature oocytes released from the PB were selected and measured for MPF activity. Consequently, the MPF of insulin-treated oocytes was significantly higher than that of EGF, IGF-1, and GH-treated oocytes. This result is similar to the nuclear maturation rate after IVM and suggests that insulin increases the MPF activity of oocytes. Although, in this study, it was not possible to determine whether the addition of insulin during IVG was associated with an increase in the MPF of SACOCs. However, it is seen as a result of the positive effects of insulin on oocyte growth indicators such as cumulus cell expansion, cytoplasmic diameter, ROS, and GSH levels.

Finally, we compared oocyte maturation and blastocyst development after PA between SAFCOCs and MAFCOCs cultured in an optimized IVG medium. It was necessary to investigate whether SAFCOCs derived from an optimized IVG environment have the developmental potential to replace MAFCOCs. Consequently, the SAFCOCs cultured in the insulin-added IVG medium had a blastocyst ratio similar to that of the MAFCOCs after PA. This demonstrated the possibility of utilizing SAFCOCs with low developmental potential, given the proper IVG culture, although an additional culture period of 2 days was required.

## 5. Conclusions

This study investigated the effects of growth factors and hormones in IVG culture on the maturation of SAFCOCs and embryonic development after PA. Insulin treatment during IVG was found to have a positive effect on nuclear maturation, embryo development after PA, and the expansion of cumulus cells. Additionally, insulin showed positive results by enhancing cytoplasmic maturation, decreasing free radical content, and increasing MPF activity. In the future, the effects of IVF on embryonic development should be explored further. The higher-quality SAFCOCs that IVF engenders could end up improving the efficiency of assisted reproductive and artificial breeding techniques.

## Figures and Tables

**Figure 1 animals-13-01206-f001:**
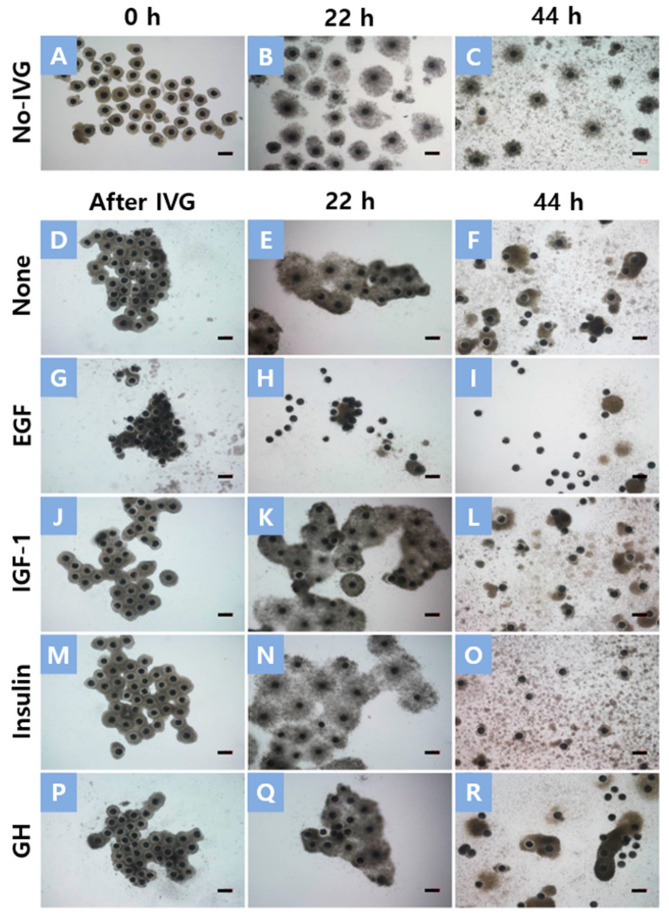
Morphology of cumulus–oocyte complexes (COCs) after in vitro growth (IVG) and in vitro maturation (IVM) culture. COCs without IVG culture (**A**) and COCs that had been cultured for 2 days in an IVG medium supplemented with None (**D**), epidermal growth factor (EGF) (**G**), insulin-like growth factor-1 (IGF-1) (**J**), insulin (**M**), and growth hormone (GH) (**P**). No-IVG COCs and IVG COCs were cultured for 22 h in an IVM medium with FSH and human chorionic gonadotrophin (**B**,**E**,**H**,**K**,**N**,**Q**) and then cultured for an additional 22 h in an FSH and human chorionic gonadotrophin-free medium (**C**,**F**,**I**,**L**,**O**,**R**). Scale bar = 200 μm.

**Figure 2 animals-13-01206-f002:**
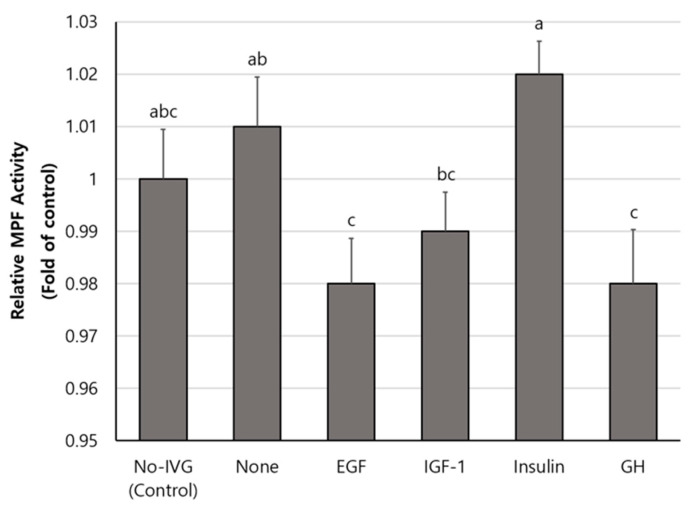
Maturation-promoting factor (MPF) activity (mean ± SEM) after in vitro maturation of small antral follicle-derived oocytes that were treated with various growth factors or hormones during in vitro growth (IVG). MPF activity was analyzed in four replicates by sampling 50 mature oocytes for each treatment. Different letters (a–c) in the bar indicate significant differences among treatment groups (*p* < 0.05).

**Table 1 animals-13-01206-t001:** Effect of growth factor and hormone treatment during in vitro growth (IVG) culture on oocyte maturation and embryonic development after the parthenogenesis (PA) of pig oocytes derived from small antral follicles.

Treatment during IVG	IVG Culture (day)	No. COCs Examined *	% of Degraded Oocytes	No. of COCs Matured	% of Oocytes Reaching MII	No. of PA Embryos Cultured	% of Embryos Developed to		No. Cells in Blastocyst
							≥2-Cell	Blastocyst	
No-IVG	0	253	4.1 ± 1.5 ^a^	243	80.0 ± 2.2 ^b^	187	81.2 ± 1.9	27.5 ± 3.4 ^c^	34.5 ± 1.7 ^b^
None	2	252	8.9 ± 3.0 ^ab^	230	65.3 ± 3.3 ^cd^	119	74.5 ± 5.4	55.8 ± 5.2 ^b^	39.7 ± 1.5 ^a^
EGF	2	253	11.6 ± 2.9 ^b^	224	41.0 ± 5.4 ^e^	91	80.1 ± 5.0	46.9 ± 3.7 ^b^	39.5 ± 2.1 ^ab^
IGF-1	2	246	10.2 ± 1.9 ^ab^	221	72.6 ± 4.2 ^bc^	133	71.3 ± 3.4	57.8 ± 4.2 ^ab^	38.2 ± 1.5 ^ab^
Insulin	2	256	6.0 ± 2.7 ^ab^	241	93.4 ± 1.0 ^a^	193	84.0 ± 1.6	69.3 ± 3.2 ^a^	41.0 ± 1.1 ^a^
GH	2	253	10.3 ± 3.1 ^ab^	228	56.3 ± 3.1 ^d^	108	71.8 ± 7.3	51.4 ± 5.4 ^b^	39.2 ± 2.0 ^ab^

* Six replicates; COCs, cumulus–oocyte complexes; EGF, epidermal growth factor; IGF-1, insulin-like growth factor-1; GH, growth hormone; ^a–e^ Values in the same column with different superscript letters are significantly different (*p* < 0.05).

**Table 2 animals-13-01206-t002:** Effect of growth factor and hormone treatment during in vitro growth (IVG) culture on cumulus cell expansion, diameter of oocytes, and size of the perivitelline space (PVS) after IVG and in vitro maturation (IVM).

Treatment during IVG	No. Oocytes Examined	Cumulus Expansion Score after IVM *	No. Oocytes Examined	Oocyte Diameter (µm) after IVG	Size of PVS (µm) after IVG	No. Oocytes Examined	Oocyte Diameter (µm) after IVM	PVS Size (µm) of MII Oocytes after IVM
No-IVG	163	1.90 ± 0.05 ^b^	75	118.9 ± 0.5 ^c^	1.9 ± 0.1 ^b^	56	110.2 ± 0.9 ^d^	3.4 ± 0.3 ^ab^
None	154	1.27 ± 0.08 ^c^	71	120.3 ± 0.5 ^ab^	2.4 ± 0.1 ^a^	42	113.2 ± 0.7 ^c^	3.4 ± 0.3 ^ab^
EGF	163	ND **	74	120.7 ± 0.4 ^ab^	2.3 ± 0.1 ^a^	43	112.8 ± 0.8 ^c^	4.0 ± 0.3 ^a^
IGF-1	156	1.92 ± 0.08 ^b^	75	120.4 ± 0.5 ^b^	2.5 ± 0.1 ^a^	45	114.2 ± 1.0 ^bc^	2.8 ± 0.3 ^bc^
Insulin	161	3.30 ± 0.06 ^a^	72	122.0 ± 0.4 ^a^	2.3 ± 0.1 ^a^	51	115.8 ± 0.7 ^ab^	2.6 ± 0.2 ^c^
GH	157	1.44 ± 0.13 ^c^	73	121.6 ± 0.5 ^ab^	2.2 ± 0.0 ^ab^	45	116.9 ± 1.0 ^a^	3.7 ± 0.3 ^a^

Three replicates; ^a–d^ Values in the same column with different superscript letters are significantly different (*p* < 0.05); * Cumulus cell expansion was scored as 0 (no response), 1 (minimum observable response with the cells in the outermost layer of the cumulus becoming round and glistening), 2 (the expansion of outer cumulus cell layers), 3 (the expansion of all cumulus cell layers except the corona radiata), and 4 (the expansion of all cumulus cell layers); ** Cumulus cell expansion was not determined because cumulus cells were detached from oocytes treated with EGF during IVG culture.

**Table 3 animals-13-01206-t003:** Effect of growth factor and hormone treatment during in vitro growth (IVG) culture on intra-oocyte glutathione (GSH) and reactive oxygen species (ROS) contents after IVM.

Treatment during IVG	IVG Culture (day)	No. of MII Oocytes Examined for GSH *	Relative Level (Pixels/Oocyte) of GSH	No. of MII Oocytes Examined for ROS **	Relative Level (Pixels/Oocyte) of ROS
No-IVG	0	58	1.00 ± 0.07 ^b^	76	1.00 ± 0.04 ^b^
None	2	50	1.10 ± 0.09 ^ab^	79	0.72 ± 0.04 ^a^
EGF	2	46	1.13 ± 0.06 ^ab^	42	0.71 ± 0.06 ^a^
IGF-1	2	67	1.22 ± 0.07 ^a^	58	0.68 ± 0.04 ^a^
Insulin	2	64	1.20 ± 0.07 ^a^	77	0.69 ± 0.04 ^a^
GH	2	48	1.24 ± 0.07 ^a^	64	0.67 ± 0.04 ^a^

* Three replicates; ^**^ Four replicates; ^a,b^ Values in the same column with different superscript letters are significantly different (*p* < 0.05); EGF, epidermal growth factor; IGF-1, insulin-like growth factor-1; GH, growth hormone.

**Table 4 animals-13-01206-t004:** Nuclear stage of small antral follicle derived pig oocytes after in vitro growth (IVG) culture in a medium supplemented with various growth factors and hormones.

Treatment during IVG	IVG Culture (day)	No. of Oocytes Examined *	Nuclear Status (%)			
			GV	MI	AI/TI	MII
No-IVG	0	75	75 (100.0 ± 0.0)	0 (0.0 ± 0.0)	0 (0.0 ± 0.0)	0 (0.0 ± 0.0)
None	2	69	65 (93.9 ± 3.1)	1 (1.7 ± 1.7)	1 (1.7 ± 1.7)	2 (2.8 ± 2.8)
EGF	2	73	67 (91.7 ± 4.2)	4 (5.6 ± 3.7)	1 (1.4 ± 1.4)	1 (1.3 ± 1.3)
IGF-1	2	72	67 (92.9 ± 3.0)	1 (1.4 ± 1.4)	2 (2.8 ± 1.4)	2 (2.9 ± 2.9)
Insulin	2	70	68 (97.2 ± 1.4)	0 (0.0 ± 0.0)	1 (1.4 ± 1.4)	1 (1.4 ± 1.4)
GH	2	72	68 (94.6 ± 3.5)	2 (2.7 ± 2.7)	1 (1.4 ± 1.4)	1 (1.3 ± 1.3)

* Three replicates; GV, germinal vesicle; MI, metaphase I; AI, anaphase I; TI, telophase I; MII, metaphase II; EGF, epidermal growth factor; IGF-1, insulin-like growth factor-1; GH, growth hormone.

**Table 5 animals-13-01206-t005:** Nuclear stage at 22 h of in vitro maturation of small antral follicle derived pig oocytes cultured in in vitro growth medium containing various growth factors or hormones.

Treatment during IVG	IVG Culture (day)	No. of Oocytes Examined *	Nuclear Status (%) *			
			GV	MI	AI/TI	MII
No-IVG	0	65	21 (32.9 ± 1.9) ^a^	36 (54.7 ± 2.5) ^a^	3 (6.5 ± 6.5)	5 (6.0 ± 6.0)
None	2	76	45 (59.5 ± 3.4) ^ab^	25 (32.2 ± 9.3) ^ab^	1 (1.4 ± 1.4)	5 (6.9 ± 4.5)
EGF	2	74	49 (59.2 ± 16.3) ^ab^	15 (25.7 ± 12.4) ^ab^	6 (7.1 ± 2.3)	4 (8.1 ± 6.2)
IGF-1	2	64	32 (49.2 ± 12.5) ^ab^	26 (41.4 ± 12.0) ^ab^	1 (1.5 ± 1.5)	5 (7.9 ± 2.1)
Insulin	2	77	29 (41.1 ± 12.5) ^ab^	37 (45.4 ± 9.7) ^ab^	1 (1.0 ± 1.0)	10 (12.5 ± 1.8)
GH	2	82	58 (70.3 ± 1.1) ^b^	19 (23.1 ± 0.1) ^b^	1 (1.9 ± 1.9)	4 (4.6 ± 0.8)

* Three replicates; ^a,b^ Values in the same column with different superscript letters are significantly different (*p* < 0.05); * GV, germinal vesicle; MI, metaphase I; AI, anaphase I; TI, telophase I; MII, metaphase II; EGF, epidermal growth factor; IGF-1, insulin-like growth factor-1; GH, growth hormone.

**Table 6 animals-13-01206-t006:** A comparison of oocyte maturation and embryonic development after parthenogenesis (PA) of small antral follicle (SAF)-derived oocytes treated with insulin during in vitro growth (IVG) culture with those of oocytes derived from medium antral follicles (MAF).

Origin of Oocytes	No. COCs Examined *	% of Degraded Oocytes	No. of COCs Matured	% of Oocytes Reaching MII	No. of PA Embryos Cultured	% of Embryos Developed to		No. Cells in Blastocyst
						≥2-cell	Blastocyst	
MAF	248	10.2 ± 2.9	223	94.3 ± 0.9 ^a^	160	88.9 ± 2.9 ^a^	51.9 ± 6.4	41.8 ± 2.0
SAF	209	14.7 ± 2.1	178	80.1 ± 4.1 ^b^	143	71.1 ± 5.1 ^b^	55.8 ± 3.7	43.1 ± 1.9

* Five replicates; COCs, cumulus-oocyte-complexes; ^a,b^ Values in the same column with different superscript letters are significantly different (*p* < 0.05).

## Data Availability

The data presented in this study are available on request from the corresponding author.

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
