# Peer review of "Effect of Growth Factors and Hormones during In Vitro Growth Culture of Cumulus-Oocyte-Complexes Derived from Small Antral Follicles in Pigs"

_animals, 2023, doi:10.3390/ani13071206_

Round 1

Reviewer 1 Report

In this article entitled “Effect of growth factors and hormones during in vitro growth culture of oocytes derived from small antral follicles in pigs” is overall a well-written and easy to read manuscript. We feel that it is pertinent research that offer valuable insight in improving the developmental competence of oocytes derived from small follicles. However, we have several comments/concerns that we feel should be addressed in order to improve your manuscript. They are listed below:

·       In your title, simple summary and abstract you repeatedly state that “oocytes” were subjected to IVG prior to IVM. While this is technically correct, “COCs” would be a more accurate description, especially since measuring cumulus expansion was one of your goals.

·       On Line 71 you talk about the goal of promoting cytoplasmic maturation, without precocious nuclear maturation. In your M&M section you state that your IVG medium contains dbcAMP, I feel that this should be mentioned in the introduction. Perhaps the fact that some research groups routinely use dbcAMP for the first 22h of IVM (similar to your IVG treatment) is worth mentioning in the discussion. Have you considered testing supplementation with dbcAMP during IVG and the first 22h of IVM together, or other meiotic inhibitors such as CNP?

·       On line 78 you talk about the use of different growth factors and hormones. If your argument is that small follicles would benefit from these additional growth factors, is there evidence to suggest that the concentrations of these are higher in follicular fluid from larger follicles? This should be discussed. If so, would you expect COCs from medium and large follicles to also benefit from these compounds? It would also help you to justify the concentrations used.

·       Line 78. It would be more accurate to say “porcine follicular fluid” rather than “porcine follicles”.

·       Line 97, you mention late embryonic development. Do you mean early embryonic development?

·       Line 104. What is “oocyte growth”? Do you mean cumulus expansion? Oocyte diameter?

·       Line 105. “Parthenogenesis (PA)” should be “parthenogenetic activation (PA)”

·       Line 131. How many COCs per well? Why did you culture them for 2 days, does that mean 48h exactly? I believe other groups have tested prolonged IVG culture periods.

·       Line 159. What does glistening cumulus mean? This seems highly subjective. Have you considered measuring the cumulus expansion directly using software? i.e. imageJ.

·       Line 173. You talk about denuded oocytes before you describe how they were denuded.

·       Line 199. How was cleavage and blastocyst rate calculated? Cleavage out of total oocyte or mature oocytes? Is blastocyst rate calculated over cleavage or oocytes? Please clarify.

·       The total number of COCs used for each experiment as well as the number of replicates for each experiment needs to be made clearer throughout the manuscript.

·       Table 1. Rather than a separate table at the end (table 6) why didn’t you include a control with medium/large follicles in table 1?

·       Table 1. Please re-assess and confirm the statistics throughout the manuscript. For the blastocyst cell count in Table 1 for example, you show that 39.5 ± 2.1 to be significantly different to 39.7 ± 1.5. With such a small sample size, and the expected large variation in cell count among blastocysts, this kind of statistical significance with a numerical difference of 0.2 seems unlikely.

·       Line 240. You say that insulin resulted in a better cumulus expansion than EGF. However, you did not measure the cumulus expansion from EGF.

·       Table 2. How many replicates? For the cumulus expansion scoring was each COC scored independently? Or was each cohort of COCs scored as a whole? Why was the EGF group not assigned a cumulus expansion score, it seems strange that the cumulus look tightly attached following IVG, but then disassociate during IVM, when the IVM conditions are the same among all groups. I know you mention this in the discussion, but it is still a strange result considering EGF is routinely used in many IVM media compositions. How many replicates showed this phenotype?

·       Table 3. The relative fluorescent for GSH and ROS is measured relative to what? If the treatments are measured relative to the no-IVG group, why does the no-IVG group have a standard error? Oocytes from how many replicates were included?

·       Table 3. How do you account for the no-IVG group having the highest ROS content? Those oocytes came directly from the follicle/abattoir (presumably very good conditions compared to those kept for 48h in vitro), while the treatments were kept for an 48h in culture (under conditions which are not ideal). I find these results strange considering there were no antioxidants supplemented and that in vitro conditions are well known to induce oxidative stress (high atmospheric oxygen, exposure to light, VOCs, etc.) compared to in vivo conditions.

·       Table 4. How many replicates?

·       Table 5. What is the significance/purpose of this table showing maturation after 22h in IVM? Maturation after the full 44h is what is most important, and this is already shown in Table 1. Consider removing.

·       Table 6. How do you justify that your results for the SAFCOC group in this experiment seem different to your results shown in table 1? Maturation 80.1 vs. 93.4, cleavage 71.1 vs 84.0, blastocyst 69.3 vs. 55.8.

·       Discussion. More citations should be added. Recent work has shown additional growth factors supplemented in combination with EGF and IGF during IVM to be beneficial (notably FGF2 and LIF) not only in PA embryos but also SCNT and IVF embryos. Please consider citing these articles as they would complement your results as they also assess cumulus expansion and meiotic maturation in porcine oocytes.

o   Yuan, Y., Spate, L. D., Redel, B. K., Tian, Y., Zhou, J., Prather, R. S., & Roberts, R. M. (2017). Quadrupling efficiency in production of genetically modified pigs through improved oocyte maturation. Proceedings of the National Academy of Sciences, 114(29), E5796-E5804.

o   Currin, L., Glanzner, W. G., Gutierrez, K., de Macedo, M. P., Guay, V., Baldassarre, H., & Bordignon, V. (2022). Optimizing swine in vitro embryo production with growth factor and antioxidant supplementation during oocyte maturation. Theriogenology, 194, 133-143.

Author Response

Reviewer 1

I would like to take this opportunity to express our thanks to the reviewer for the positive feedback and helpful comments for correction or modification. My co-authors and I very much appreciated the encouraging, critical and constructive comments on this manuscript by the reviewer. The comments have been very thorough and useful in improving the manuscript.

Comments

 In your title, simple summary and abstract you repeatedly state that “oocytes” were subjected to IVG prior to IVM. While this is technically correct, “COCs” would be a more accurate description, especially since measuring cumulus expansion was one of your goals.

Response

Following the reviewer's suggestion, we changed the title 'oocyte' to 'cumulus-oocyte-complexes'.

Line 24, 34, 36: ‘oocytes’ to ‘Cumulus-oocyte-complexes’ or ‘COCs’

Comments

On Line 71 you talk about the goal of promoting cytoplasmic maturation, without precocious nuclear maturation. In your M&M section you state that your IVG medium contains dbcAMP, I feel that this should be mentioned in the introduction. Perhaps the fact that some research groups routinely use dbcAMP for the first 22h of IVM (similar to your IVG treatment) is worth mentioning in the discussion. Have you considered testing supplementation with dbcAMP during IVG and the first 22h of IVM together, or other meiotic inhibitors such as CNP?

Response

Thanks for the nice comments. However, since this study focused on the IVG process, CNP treatment during the initial 22 hours of IVM was not considered. Based on the reviewer's comment, we have added the sentence below to the introductory and discussion sessions.

Line 76-81: ‘Cyclic adenosine monophosphate (cAMP) plays an important role in inhibiting meiotic resumption [7]. Furthermore, maintaining gap junction communication between oocytes and cumulus cells during IVM of immature oocytes is critical for oocyte cytoplasmic maturation [8]. In pigs, the IVG culture system, using a medium containing dibutyryl cAMP (dbcAMP), induces oocyte growth while inhibiting spontaneous meiotic resumption, and has been widely used to produce mature oocytes.’

Line 429-431: Previous studies have reported that adding 1 mM dbcAMP to cultures arrests porcine SAFCOCs in the GV phase for more than 5 days [35].

Comments

On line 78 you talk about the use of different growth factors and hormones. If your argument is that small follicles would benefit from these additional growth factors, is there evidence to suggest that the concentrations of these are higher in follicular fluid from larger follicles? This should be discussed. If so, would you expect COCs from medium and large follicles to also benefit from these compounds? It would also help you to justify the concentrations used.

Response

Quantitative analysis of growth factors and hormones in follicular fluid according to the size of pig follicles is not sufficient until now. Therefore, this study was designed for IVG of small follicle-derived COCs based on effective substances and optimal concentrations in the IVM process of medium-sized follicle-derived COCs. We have added references on IGF-1 and GH, which have been shown to be correlated with follicular size, to the manuscript as follows.

Line 97-99: Oberlender et al. reported that follicular fluid derived from small follicles (2-5 mm) contained a lower concentration of IGF-1 compared to follicular fluid derived from large follicles (6-10 mm) [20].

Line 102-107: It has been reported that GH acts directly on ovarian follicles, and that the action of GH depends on the size of the follicle, showing more effects on cells of large follicles than on small-sized follicles [24].

Comments

Line 78. It would be more accurate to say “porcine follicular fluid” rather than “porcine follicles”.

Response

Line 91: It was changed to 'porcine follicular fluid' according to the reviewer's suggestion.

Comments

Line 97, you mention late embryonic development. Do you mean early embryonic development?

Response

Line 117: It was changed to ‘early embryonic development’ according to the reviewer's suggestion.

Comments

Line 104. What is “oocyte growth”? Do you mean cumulus expansion? Oocyte diameter?

Response

Ambiguous phrasing has been corrected to clarify phrasing.

Line 124: ‘oocyte growth’ to ‘oocyte size’

Comments

Line 105. “Parthenogenesis (PA)” should be “parthenogenetic activation (PA)”

Response

Line 125: It was changed to ‘parthenogenetic activation (PA)’ according to the reviewer's suggestion.

Comments

Line 131. How many COCs per well? Why did you culture them for 2 days, does that mean 48h exactly? I believe other groups have tested prolonged IVG culture periods.

Response

The optimal culture period for IVG was determined by our group's other experimental results (unpublished data). Based on the reviewer's comment, we have added the sentence below to the Materials and Methods sessions.

Line 150: ‘Forty to fifty SAFCOCs were cultured for 2 days (48h) in 500 μL of IVG medium~’

Comments

Line 159. What does glistening cumulus mean? This seems highly subjective. Have you considered measuring the cumulus expansion directly using software? i.e. imageJ.

Response

In this study, we did not consider measuring cumulus expansion using software. The degree of expansion of cumulus cells was visually observed and scored by applying the criteria presented in the report by Barbara C et al., 1990. 'Glistening' is a phenomenon in which expanded cumulus cells shine when viewed under a microscope, and can be easily distinguished from unexpanded dark cumulus cells with the naked eye.

While we agree with the reviewer's comment that the measuring method is subjective, we have tried to rule out subjectivity. The images of the expanded cumulus cells were scored by 4 researchers without bias, and the average result of each researcher was calculated.

Comments

Line 173. You talk about denuded oocytes before you describe how they were denuded.

Response

Line 201: ‘After IVG or IVM, mechanical pipetting was performed while the oocytes were kept in 0.1% hyaluronidase because of the sticky character of the cumulus cells.’

Comments

Line 199. How was cleavage and blastocyst rate calculated? Cleavage out of total oocyte or mature oocytes? Is blastocyst rate calculated over cleavage or oocytes? Please clarify.

Response

Line 237: ‘Cleavage and blastocyst formation (based on the number of PA embryo cultured) were evaluated on Days 2 and 7, respectively, with the day of PA designated as Day 0.’

Comments

The total number of COCs used for each experiment as well as the number of replicates for each experiment needs to be made clearer throughout the manuscript.

Response

Following the reviewer's suggestion, we added the total number of COCs used in the experiment to the results section and table 1 and 6.

Line 261: ‘For this experiment, total 1513 SAFCOCs were used.’

Line 261: ‘For this experiment, total 248 MAFCOCs and 209 SAFCOCs were used.’

Comments

Table 1. Rather than a separate table at the end (table 6) why didn’t you include a control with medium/large follicles in table 1?

Response

The main purpose of this study was to investigate the effects of various growth factors and hormones in the IVG medium of SAFCOCs. Therefore, most of the experiments were performed with SAFCOCs without MAFCOCs treatment. Table 6 was conducted to investigate whether SAFCOCs derived from an optimized IVG culture environment could replace MAFCOC in assisted reproductive technology. We modified the discussion session to highlight these points.

Line 473-475: ‘It was necessary to investigate whether SAFCOCs derived from an optimized IVG environment have the developmental potential to replace MAFCOCs.’

Comments

Table 1. Please re-assess and confirm the statistics throughout the manuscript. For the blastocyst cell count in Table 1 for example, you show that 39.5 ± 2.1 to be significantly different to 39.7 ± 1.5. With such a small sample size, and the expected large variation in cell count among blastocysts, this kind of statistical significance with a numerical difference of 0.2 seems unlikely.

Response

In response to the reviewer's suggestion, we reanalyzed the statistics of all results and found a mistake in the notation of significant differences in Table 1. We modified the results session and table 1. Thank you.

Comments

Line 240. You say that insulin resulted in a better cumulus expansion than EGF. However, you did not measure the cumulus expansion from EGF.

Response

We deleted ‘EGF’ from the sentence. (Line 284)

Comments

Table 2. How many replicates? For the cumulus expansion scoring was each COC scored independently? Or was each cohort of COCs scored as a whole? Why was the EGF group not assigned a cumulus expansion score, it seems strange that the cumulus look tightly attached following IVG, but then disassociate during IVM, when the IVM conditions are the same among all groups. I know you mention this in the discussion, but it is still a strange result considering EGF is routinely used in many IVM media compositions. How many replicates showed this phenotype?

Response

The cumulus cell expansion assay was repeated a total of 4 times and cumulus expansion scores were calculated for individual COCs. The sentence below is attached for clarity.

Line 185: The cumulus expansion scores were calculated for individual COCs.

In the case of the EGF-treated group, many cumulus cells expanded and were separated from the oocyte and attached to the bottom after IVG. We observed a phenotype in which cumulus cells were separated from oocytes in all experiments where EGF was used. This phenomenon was also observed when PVA was used instead of follicular fluid in the culture medium during the IVM process. In this study, PVA was used instead of follicular fluid in the culture medium for defined medium conditions during IVG. It can be assumed that the use of PVA during IVG was responsible for the separation between the expanded cumulus cells and oocytes after IVM.

We added this assumption to the discussion session.

Line 409-415: many expanded cumulus cells and were separated from the oocyte and attached to the bottom in the case of the EGF-treated group after IVG. This phenomenon was also observed when PVA was used instead of follicular fluid in the culture medium during the porcine IVM process. In this study, PVA was used in the culture medium for defined medium conditions during IVG. It can be assumed that the use of PVA during IVG was responsible for the separation between the expanded cumulus cells and oocytes after IVM.

Line 417-419: in our IVG culture conditions, addition of EGF caused premature cumulus expansion and stimulated the separation of cumulus cells from oocytes.

Comments

Table 3. The relative fluorescent for GSH and ROS is measured relative to what? If the treatments are measured relative to the no-IVG group, why does the no-IVG group have a standard error? Oocytes from how many replicates were included?

Response

ROS or GSH normalized the mean fluorescence intensity of each oocyte to 1 in the No-IVG group. Therefore, the standard error of the IVG group was derived from the difference in the value of each oocyte.

In the case of number of cells in blastocyst, cumulus expansion score, oocyte dimeter, size of PVS, GSH, and ROS analysis, statistical analysis was performed with one oocyte or one blastocyst set as one replicate. That is, the number of oocytes or blastocysts used in the experiment is equal to the number of replicates. Therefore, replicates are not indicated in the table legend.

Comments

Table 3. How do you account for the no-IVG group having the highest ROS content? Those oocytes came directly from the follicle/abattoir (presumably very good conditions compared to those kept for 48h in vitro), while the treatments were kept for an 48h in culture (under conditions which are not ideal). I find these results strange considering there were no antioxidants supplemented and that in vitro conditions are well known to induce oxidative stress (high atmospheric oxygen, exposure to light, VOCs, etc.) compared to in vivo conditions.

Response

In this study, all oocytes were measured for ROS after IVM. It is possible that IVG-derived oocytes had a sufficient cytoplasmic maturation period and had a higher ability to defend against oxidative stress than No-IVG oocytes. Intracellular GSH protects oocyte from proteasome inhibition-induced oxidative stress. In our results, oocytes with low levels of ROS were observed in oocytes with high levels of GSH. This result can be assumed that IVG improved cytoplasmic maturation and intracytoplasmic GSH concentration of oocytes.

Comments

Table 4. How many replicates?

Response

Three replicates. Added ‘replicates’ to table legend.

Comments

Table 5. What is the significance/purpose of this table showing maturation after 22h in IVM? Maturation after the full 44h is what is most important, and this is already shown in Table 1. Consider removing.

Response

The reason for analyzing the nucleus of oocytes at 22 hours after IVM was to investigate whether dbcAMP, growth factors, and hormones treated during IVG delayed oocyte meiotic resumption after the start of IVM.

Comments

Table 6. How do you justify that your results for the SAFCOC group in this experiment seem different to your results shown in table 1? Maturation 80.1 vs. 93.4, cleavage 71.1 vs 84.0, blastocyst 69.3 vs. 55.8.

Response

The authors also acknowledge that there are numerical differences in the results of SAFCOC in Tables 1 and 6. This study was conducted over a long period of time (more than 6 months), and there were some differences in ovarian quality depending on when the experiment was conducted. Nevertheless, in Table 6, the two treatments were compared under the same conditions except for the size of the follicle from which the oocytes were collected.

Comments

Discussion. More citations should be added. Recent work has shown additional growth factors supplemented in combination with EGF and IGF during IVM to be beneficial (notably FGF2 and LIF) not only in PA embryos but also SCNT and IVF embryos. Please consider citing these articles as they would complement your results as they also assess cumulus expansion and meiotic maturation in porcine oocytes.

Yuan, Y., Spate, L. D., Redel, B. K., Tian, Y., Zhou, J., Prather, R. S., & Roberts, R. M. (2017). Quadrupling efficiency in production of genetically modified pigs through improved oocyte maturation. Proceedings of the National Academy of Sciences, 114(29), E5796-E5804.

Currin, L., Glanzner, W. G., Gutierrez, K., de Macedo, M. P., Guay, V., Baldassarre, H., & Bordignon, V. (2022). Optimizing swine in vitro embryo production with growth factor and antioxidant supplementation during oocyte maturation. Theriogenology, 194, 133-143. 

Response

At the suggestion of the reviewer, we cited the above two references in the discussion session.

Line 381-390: ‘Recent work has shown additional growth factors supplemented in combination with EGF and IGF during IVM to be beneficial (notably fibroblast growth factor 2 and leukemia inhibitory factor) not only in PA embryos but also SCNT and IVF embryos [36,37]. How-ever, growth factor and hormone requirements in the in vitro environment may vary de-pending on the developmental period and origin of the oocyte.’

Thank you.

Reviewer 2 Report

The manuscript reports on studies to examine the feasibility of using follicles less than 3mm in diameter from slaughterhouse pigs as the source for oocytes. The manuscript is well-written and demonstrates that the addition of insulin to culture media will enhance oocyte competence.

Author Response

Reviewer 2

I would like to take this opportunity to express our thanks to the reviewer for the positive feedback and helpful comments for correction or modification. My co-authors and I very much appreciated the encouraging, critical and constructive comments on this manuscript by the reviewer. The comments have been very thorough and useful in improving the manuscript.

Comments

The manuscript reports on studies to examine the feasibility of using follicles less than 3mm in diameter from slaughterhouse pigs as the source for oocytes. The manuscript is well-written and demonstrates that the addition of insulin to culture media will enhance oocyte competence.

Response

Thank you for the reviewer's good evaluation.

Reviewer 3 Report

The presented paper is interesting and well structured, however before publication Authors should consider the following comments and suggestions:

Experimental design: in each paragraph please add the total number (n) of oocytes used, then please add info on number of (n) oocytes used in each stage of experiment and provide the number of replicates per each experiment.

Materials and methods

2.7.  please  add a brief methodology for measurement of oocyte diameter

2.8. how was the presence of PB I assessed, have the Authors only inspect the oocyte or used manipulation e.q. rolling to evaluate each surface of the cell?

Please specify the criteria for the number of cells in the blastocysts, were the counts done manually and what was the limit of the number of cells to qualify the embryo as a blastocyst? Please provide a reference to this, as later in the results Authors count embryos containing 34 cells as blastocysts, which seems a bit of a stretch...

2.9 please add 'Maturation Promoting Factor' (MPF) activity assay in title

Table 4 and 5 – please unify and show data as percentage or numbers (or both).

Results

Tables (all) – add information on the initial number of oocytes collected for each stage of experiment and in each treatment group.

Would it be possible to provide any pictures of fluorescent staining results, either oocyte nuclear status, blastocyst cell count, GSH, ROS or MPF concentration?

Discussion

From line 355 onwards (up to 381) – are there available any published research on this subject to compare and discuss your results with?

Author Response

Reviewer 3

I would like to take this opportunity to express our thanks to the reviewer for the positive feedback and helpful comments for correction or modification. My co-authors and I very much appreciated the encouraging, critical and constructive comments on this manuscript by the reviewer. The comments have been very thorough and useful in improving the manuscript.

Comments

The presented paper is interesting and well structured, however before publication Authors should consider the following comments and suggestions:

Experimental design: in each paragraph please add the total number (n) of oocytes used, then please add info on number of (n) oocytes used in each stage of experiment and provide the number of replicates per each experiment.

Response

Thank you for the reviewer's kind comments. We supplemented information on the total number of oocytes used and the number of replicates per experiment based on comments from reviewers.

Comments

2.7.  please  add a brief methodology for measurement of oocyte diameter

Response

As suggested by the reviewer, we have attached Figure S2.

Comments

2.8. how was the presence of PB I assessed, have the Authors only inspect the oocyte or used manipulation e.q. rolling to evaluate each surface of the cell?

Response

Line 228-229: ‘Each oocyte was rolled with a mouse pipette under a microscope, and only oocytes that extruded the first polar body (PB) were selected and allocated to PA.’

Comments

Please specify the criteria for the number of cells in the blastocysts, were the counts done manually and what was the limit of the number of cells to qualify the embryo as a blastocyst? Please provide a reference to this, as later in the results Authors count embryos containing 34 cells as blastocysts, which seems a bit of a stretch...

Response

Line 239-240: Only embryos with 18 or more stained nuclei were recognized as blastocysts.

Comments

2.9 please add 'Maturation Promoting Factor' (MPF) activity assay in title

Response

We did not write the full name in the title because we put the MPF's full name on line 172.

Comments

Table 4 and 5 – please unify and show data as percentage or numbers (or both).

Response

We have indicated the numbers and percentages in Tables 4 and 5.

Comments

Tables (all) – add information on the initial number of oocytes collected for each stage of experiment and in each treatment group.

Response

We added the total number of COCs used in each experiment to Tables 1 and 6.

Comments

Would it be possible to provide any pictures of fluorescent staining results, either oocyte nuclear status, blastocyst cell count, GSH, ROS or MPF concentration?

Response

A representative image of the nuclear status was added as Figure S1. However, we apologize for not being able to attach images of blastocyst cell count, GSH, ROS, or MPF because there are no representative suitable images.

Comments

From line 355 onwards (up to 381) – are there available any published research on this subject to compare and discuss your results with?

Response

The authors complemented the discussion session with citations of published results that were comparable to our results.

Line 429-431: ‘Previous studies have reported that adding 1 mM dbcAMP to cultures arrests porcine SAFCOCs in the GV phase for more than 5 days [35].’

Line 436-439: ‘It has been reported that GH receptors are localized in follicular cells in most species (rat [48], bovine [49], porcine [50]). Therefore, it seems that the addition of GH in the IVG culture could not have a positive effect on the growth and maturation of oocytes.’

Thank you.

Round 2

Reviewer 3 Report

I greatly appreciate the substantial improvement the authors done on the manuscript.

However, I cannot still accept the assumption on the number of blastomeres in blastocyst. Please consider the following (Davis DL: Culture and storage of pig embryos. J. Reprod. Fen., Suppl. 33 (1985), 115-124):

‘Cleavage and blastocyst stages of pig embryos: Day 2: 2-cell; Day 3: 4-cell; Day 5: Morula 16-30 cells; Day 6: at least 60 cells, Day 7: Hatching blastocyst 175 cells’

Most of researchers define blastocysts as embryos >30-50 cells with blastocoel cavity formation, the latter being the crucial condition (e.g. Herrick et al. 2007, Sananmuang et al. 2011). Embryos with lower number of cells are sometimes classified as:  pre-morula (2–15 cells), early (16–32), mid- (33–50), or late (>50) morula or early (50–100), mid- (101–150), late (51–200) or expanding (>200) blastocyst. Furthermore, the exact stage of embryo development is observed on particular day of embryo culture. Authors performed their observations on day 2 and 7 of embryo culture. The 2nd day inspection would allow them to confirm the initial cleavages, while the second inspection (on Day 7) should show blastocyst (containing at least 60 cells),  or earlier stages of embryos experiencing  developmental blockage or degeneration. In my opinion it cannot be stated that 16-cell embryo on day 7 was a blastocyst.  The term blastocyst should be changed into different term (‘embryos > 16 cells’, ‘pre-morula embryos’ or similar).

Is the term ‘mouse pipette’ accurate? Would it be rather ‘mouth pipette’?

Author Response

Comments

I greatly appreciate the substantial improvement the authors done on the manuscript.

However, I cannot still accept the assumption on the number of blastomeres in blastocyst. Please consider the following (Davis DL: Culture and storage of pig embryos. J. Reprod. Fen., Suppl. 33 (1985), 115-124):

‘Cleavage and blastocyst stages of pig embryos: Day 2: 2-cell; Day 3: 4-cell; Day 5: Morula 16-30 cells; Day 6: at least 60 cells, Day 7: Hatching blastocyst 175 cells’

Most of researchers define blastocysts as embryos >30-50 cells with blastocoel cavity formation, the latter being the crucial condition (e.g. Herrick et al. 2007, Sananmuang et al. 2011). Embryos with lower number of cells are sometimes classified as:  pre-morula (2–15 cells), early (16–32), mid- (33–50), or late (>50) morula or early (50–100), mid- (101–150), late (51–200) or expanding (>200) blastocyst. Furthermore, the exact stage of embryo development is observed on particular day of embryo culture. Authors performed their observations on day 2 and 7 of embryo culture. The 2nd day inspection would allow them to confirm the initial cleavages, while the second inspection (on Day 7) should show blastocyst (containing at least 60 cells), or earlier stages of embryos experiencing developmental blockage or degeneration. In my opinion it cannot be stated that 16-cell embryo on day 7 was a blastocyst.  The term blastocyst should be changed into different term (‘embryos > 16 cells’, ‘pre-morula embryos’ or similar).

Response

We thank the reviewer for her thoughtful review of our work and kind words. However, we do not agree with the reviewer's comments.

The number of blastocysts varies greatly depending on the animal species, origin of the oocyte or embryo (in vivo or in vitro), in vitro embryo production methods (e.g. in vitro fertilization, parthenogenesis, somatic cell nuclear transfer, etc.), and in vitro culture conditions (e.g. whether serum is added or not, nutrients added, etc.).

The cell number of pig embryos presented in the first paper presented by the reviewer ‘Davis DL: Culture and storage of pig embryos. J. Reprod. Fen., Suppl. 33 (1985), 115-124’ was embryos derived from naturally fertilized. Embryos generated through natural mating have a much higher number of blastocyst cells than embryos produced in vitro. The number of cells in the blastocyst presented in Davis' study may show a significant difference from our study.

In the case of the second and third references presented by the reviewer (only the author's name and year information is provided, it is impossible to find the exact paper), if the paper below is correct, there is a difference in species. These two papers are the results of studies in feline embryos, in which case the number of blastocysts may be much higher than in pigs.

Herrick JR, Bond JB, Magarey GM, Bateman HL, Krisher RL, Dunford SA, Swanson WF. Toward a feline-optimized culture medium: impact of ions, carbohydrates, essential amino acids, vitamins, and serum on development and metabolism of in vitro fertilization-derived feline embryos relative to embryos grown in vivo. Biol Reprod. 2007 May;76(5):858-70. doi: 10.1095/biolreprod.106.058065. Epub 2007 Jan 31. PMID: 17267698.

Sananmuang T, Tharasanit T, Nguyen C, Phutikanit N, Techakumphu M. Culture medium and embryo density influence on developmental competence and gene expression of cat embryos. Theriogenology. 2011 Jun;75(9):1708-19. doi: 10.1016/j.theriogenology.2011.01.008. Epub 2011 Mar 11. PMID: 21396699.

We present the results of studies that can confirm the cell number of pig-derived blastocysts that have recently been published (in the last 10 years) as follows.

  1. 26~45 cells in blastocyst / SCNT / pig

Jiao D, Cheng W, Zhang X, Zhang Y, Guo J, Li Z, Shi D, Xiong Z, Qing Y, Jamal MA, Xu K, Zhao HY, Wei HJ. Improving porcine SCNT efficiency by selecting donor cells size. Cell Cycle. 2021 Nov;20(21):2264-2277. Doi: 10.1080/15384101.2021.1980983. Epub 2021 Sep 29. PMID: 34583621; PMCID: PMC8794526.

  1. 22~37 cells in blastocyst / parthenogenesis / pig

Chen Q, Gao L, Li J, Yuan Y, Wang R, Tian Y, Lei A. α-Ketoglutarate Improves Meiotic Maturation of Porcine Oocytes and Promotes the Development of PA Embryos, Potentially by Reducing Oxidative Stress through the Nrf2 Pathway. Oxid Med Cell Longev. 2022 Feb 21;2022:7113793. doi: 10.1155/2022/7113793. PMID: 35237383; PMCID: PMC8885182.

  1. 28~38 cells in blastocyst / parthenogenesis / pig

Lee MH, Jeong PS, Sim BW, Kang HG, Kim MJ, Lee S, Yoon SB, Kang P, Park YH, Kim JS, Song BS, Koo DB, Kim SU. Induction of autophagy protects against extreme hypoxia-induced damage in porcine embryo. Reproduction. 2021 Apr;161(4):353-363. doi: 10.1530/REP-20-0311. PMID: 33528381.

  1. 22~38 cells in blastocyst / parthenogenesis / pig

Li J, Wang R, Chen Q, Tian Y, Gao L, Lei A. Salidroside improves porcine oocyte maturation and subsequent embryonic development by promoting lipid metabolism. Theriogenology. 2022 Oct 15;192:89-96. doi: 10.1016/j.theriogenology.2022.08.028. Epub 2022 Aug 29. PMID: 36084388.

  1. 23~52 cells in blastocyst / parthenogenesis / pig

Oi A, Tasaki H, Munakata Y, Shirasuna K, Kuwayama T, Iwata H. Effects of reaggregated granulosa cells and oocytes derived from early antral follicles on the properties of oocytes grown in vitro. J Reprod Dev. 2015;61(3):191-7. doi: 10.1262/jrd.2014-123. Epub 2015 Feb 19. PMID: 25740588; PMCID: PMC4498376.

  1. “Compaction of blastomeres was seen in all embryos in this study including one with as few as 8 cells and these were morphologically classified as morulae. Blastocoel formation usually occurred after the 4th cleavage division (16 cells or more) although three embryos with 9, 12 and 15 total cells contained cavities. About half the embryos with 17 –32 cells contained a cavity as did nearly all embryos with a greater total cell number.”

Papaioannou VE, Ebert KM. The preimplantation pig embryo: cell number and allocation to trophectoderm and inner cell mass of the blastocyst in vivo and in vitro. Development. 1988 Apr;102(4):793-803. doi: 10.1242/dev.102.4.793. PMID: 3168789.

Others

We made a mistake in the 'mouse' notation.

Corrected by 'mouth'

Thank you.
